# IMPROVING THE ACCURACY OF NEURAL NETWORKS IN ANALOG COMPUTING-IN-MEMORY SYSTEMS BY A GENERALIZED QUANTIZATION METHOD

## ABSTRACT

Crossbar-enabled analog computing-in-memory (CACIM) systems can significantly improve the computation speed and energy efficiency of deep neural networks (DNNs). However, the transition of DNN from the digital systems to CACIM systems usually reduces its accuracy. The major issue is that the weights of DNN are stored and calculated directly on analog quantities in CACIM systems. The variation and programming overhead of the analog weight limit the precision. Therefore, a suitable quantization algorithm is important when deploying a DNN into CACIM systems to obtain less accuracy loss. The analog weight has its unique advantages when doing quantization. Because there is no encoding and decoding process, the set of quanta will not affect the computing process. Therefore, a generalized quantization method that does not constrain the range of quanta and can obtain less quantization error will be effective in CACIM systems. For the first time, we introduced a generalized quantization method into CACIM systems and showed superior performance on a series of computer vision tasks, such as image classification, object detection, and semantic segmentation. Using the generalized quantization method, the DNN with 8-level analog weights can outperform the 32-bit networks. With fewer levels, the generalized quantization method can obtain less accuracy loss than other uniform quantization methods.

## 1 INTRODUCTION

Deep neural networks (DNNs) have been widely used in a variety of fields, such as computer vision (Krizhevsky et al., 2012; Simonyan & Zisserman, 2015; He et al., 2016), speech recognition (Graves et al., 2013; Hinton et al., 2012; Graves & Jaitly, 2014), natural language processing (Kim, 2014; Yang et al., 2016; Lai et al., 2015) and so on (Mnih et al., 2015; Silver et al., 2016). However, the high complexity of DNN models makes them hard to be applied on edge devices (mobile phones, onboard computers, smart sensors, wearable devices, etc.), which can only provide limited computing speed and power (Sze et al., 2017).

Crossbar-enabled analog computing-in-memory (CACIM) systems is a promising approach to facilitate the applications of DNN on edge devices (Yang et al., 2013). It can carry out some typical operations in situ, exactly where the data are located (Ielmini & Wong, 2018). Such as the multiply-accumulate operation (MAC), which is the most frequently performed operation in DNNs. The cost of data transferring for doing the operations can be reduced. Both the computation speed and energy efficiency can be improved significantly (Yao et al., 2020).

The footstone of CACIM systems for DNN is the crossbar array of the computational memory units (Hu et al., 2012). As shown in Figure 1, taking the memristor device as an example, each weight ($W_{ij}$) of the connection in one layer of a neural network is stored as the conductance state ($G_{ij}$) of a memristor. The input data are represented as the voltage ($V_i$). After applying the voltage ($V_i$) to each row, the current ($I_j$) collected at each column is exactly the MAC result according to Kirchhoff's law and Ohm's law, $I_j = \sum_i V_i G_{ij}$.

Before applying a DNN in CACIM systems, an essential step is writing the weights of DNN into the memory units, which is usually called as mapping. However, the mapping overhead is directly

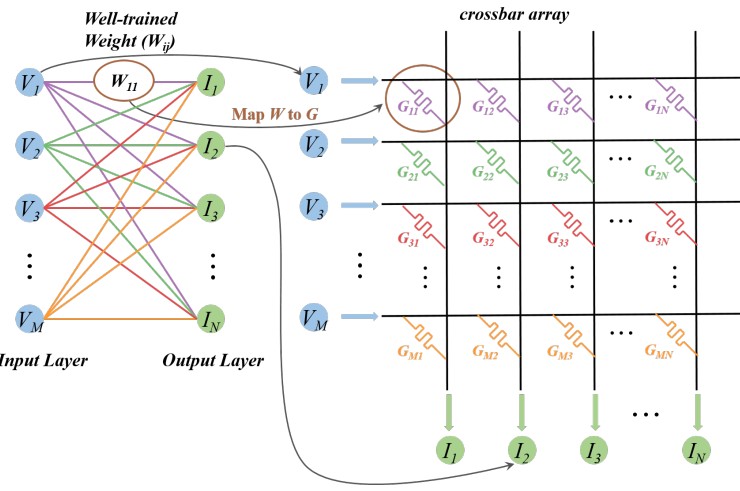

Figure 1: Schematic illustration of one layer of neural network performed in a crossbar array.

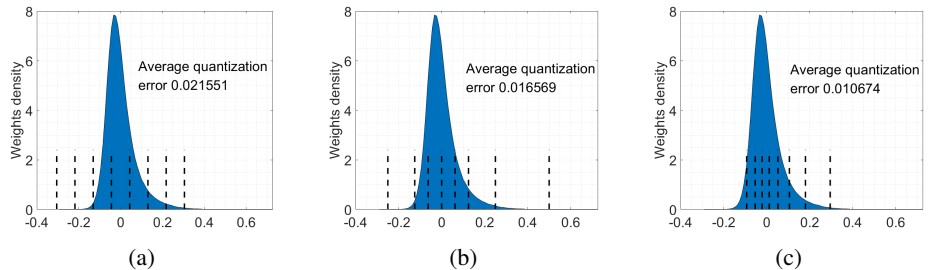

(a)                              (b)                              (c)

Figure 2: The example results of a uniform LAQ (Hou & Kwok, 2018) quantizer (a), a non-uniform (log-wise) INQ (Zhou et al., 2017) quantizer (b), and a generalized (Lloyd's) quantizer (c). We quantized the last fully connected layer's weight of ResNet-18 with 8-level quantizers. The generalized quantizer can obtain less quantization error.

related to the precision of weights.Therefore, a weight quantization method that compresses high-precision weights in DNN is important for an efficient implementation of DNN in CACIM systems.

The most important criterion of a weight quantization method is the accuracy loss, which has a strong correlation with the quantization error. The quantization error is determined by the quantizer used in the quantization method. As far as our knowledge is concerned, the generalized quantizer has not been used to quantize DNN weights in CACIM systems. The previous work used either uniform quantizers (Jung et al., 2019; Yang et al., 2019) or non-uniform quantizers (Tang et al., 2017; Sun et al., 2018). However, an empirical study has shown that the weights in one layer of DNN usually follow a bell-shape and long-tailed distribution (Han et al., 2016). Meanwhile, we found that weights in the last fully-connected layer of DNN for the classification tasks usually follow an asymmetric distribution. The generalized density-aware quantizer (GDQ), which means the quantization results are determined all depends on the data without any constraints, can obtain less quantization error than either uniform or non-uniform quantizer (Figure 2). Since the weights are stored and operated as analog quantities in CACIM systems, using GDQ to quantize the weights won't produce extra cost in the inference phase.

In CACIM systems, the noise of analog weights is inevitable and can not be ignored. The perturbations of weights will degrade the performance of networks severely. It is better to quantize the weights and improve the robustness to noise in the training process together.

In this work, we introduced a generalized density-aware quantization method and noise-aware training scheme (NATS) for DNN in CACIM systems and achieved no degradation of performance by using 8-level weights on a series of computer vision tasks. Under the same weight level, our proposed method performed better than others.

## 2 PRELIMINARY

A quantization method for DNNs consists of two parts, one is the quantizer, the other is the quantization algorithm which describes how to use the quantizer in a neural network.

### 2.1 QUANTIZER

**Formulation:** A quantizer is a function, $f : \mathbb{R} \rightarrow q$, where $q = \{q_i \in \mathbb{R} | i = 1, 2, \cdots, v\}$. $x = \{x_i \in \mathbb{R} | i = 1, 2, \cdots, d\}$ is the data to be quantized. Each $q_i$ has a corresponding domain $Q_i \subset \mathbb{R}$ that

$$f(x) = q_i, \text{ if } x \in Q_i, \tag{1}$$

where $\bigcup_{i=1}^{v} Q_i = \mathbb{R}$, and $Q_i \cap Q_j = \emptyset$ when $i \neq j$. In most cases, $\{Q_i | i = 1, 2, \cdots, v\}$ are $v$ intervals separated by $v - 1$ endpoints $e = \{e_i \in \mathbb{R} | i = 1, 2, \cdots, v - 1\}$ on the real axis. Without loss of generality, we assume that $q_1 < q_2 < \cdots < q_v$ and $e_1 < e_2 < \cdots < e_{v-1}$, that is,

$$\begin{aligned}
Q_1 &= \{x : -\infty < x \leq e_1\}, \\
Q_2 &= \{x : e_1 < x \leq e_2\}, \\
&\vdots \\
Q_{v-1} &= \{x : e_{v-2} < x \leq e_{v-1}\}, \\
Q_v &= \{x : e_{v-1} < x < \infty\}.
\end{aligned} \tag{2}$$

We use $\mathbf{\Theta} = \{q, e\}$ to denote a quantizer, and call $v = |q|$ the precision of the quantizer. The quantization error of a data point $z(x_i)$ is defined as

$$z(x_i) = f(x_i) - x_i = q_\alpha - x_i, \text{ if } x_i \in Q_\alpha. \tag{3}$$

A quantizer $\mathbf{\Theta} = \{q, e\}$ is uniform if $q$ is an arithmetic progression. The quantization method using uniform quantizer is referred as a uniform quantization method, such as the BinaryConnect (Courbariaux et al., 2015), binary weight network (BWN) (Rastegari et al., 2016), which have two levels, the ternary weight networks (TWN) (Li et al., 2016) and the trained ternary quantization (TTQ) (Zhu et al., 2016), which have three levels, and some other methods that have more levels (He & Fan, 2019; Jung et al., 2019; Yang et al., 2019; Shin et al., 2017; Esser et al., 2019).

The $q$ in a non-uniform quantizer is constrained to be a kind of non-uniform distribution. Such as $q$ is a geometric sequence (Li et al., 2020; Miyashita et al., 2016; Zhou et al., 2017). These quantization methods work best when the data to be quantized follows the corresponding exponential distribution. The work (Choi et al., 2016) adopts a $k$-means like algorithm to quantized the weights.

Beyond the uniform quantization and non-uniform quantization, the generalized quantization methods do not constrain the distribution of $q$. $q$ is determined based on the distribution of the data to be quantized, which is robust to all kinds of data distribution.

### 2.2 QUANTIZATION ALGORITHM

To accelerate the inference process of a neural network, the quantizer is usually applied to both the weights and feature maps. In most of CACIM systems, the activations are still implemented by digital circuits, which is significantly different from the analog weights. So in this work, we focused on the weight quantization problem. There are two main strategies for weight quantization. The first one directly quantizes the weights in a trained neural network without fine-tuning or retraining (Zhao et al., 2019; Banner et al., 2019). This strategy is fast and convenient, but its accuracy loss is usually greater than the other one, which will repeat the training and quantization iteratively until the performance is better enough (Courbariaux et al., 2015; Rastegari et al., 2016; Li et al., 2016; Zhu et al., 2016). In the iterative quantization scheme, starts from the pre-trained neural network is more likely to get better performance than starts from scratch (Yang et al., 2019).

# 3 GENERAL QUANTIZATION ALGORITHMS FOR DNNs IN CACIM SYSTEMS

## 3.1 LLOYD'S QUANTIZER

We used Lloyd (1982)'s quantizer to quantize the weights of DNN in this work. The $\Theta = \{q, e\}$ is a quantizer as defined in Section 2.1. Quantization distortion $E$ of a quantizer is defined as

$$E = \int_{-\infty}^{+\infty} z^2(x)\, dF(x), \tag{4}$$

$$= \sum_{\alpha=1}^{v} \int_{Q_\alpha} (q_\alpha - x)^2\, dF(x), \tag{5}$$

where $F(x)$ is the cumulative probability distribution function of $x$. To minimize $E$, the quantizer iteratively optimizes the $q$ and $e$ until the relative error of $E$ of two consecutive iterations is smaller than a given threshold.

## 3.2 NOISE-AWARE TRAINING SCHEME

We used the noise-aware training scheme (Murray & Edwards, 1994) to improve the robustness to weight noise in this work. A Gaussian noise with zero mean and $\sigma$ standard deviation was added to each weight when doing the forward calculation. The $\sigma$ is determined by the production of the maximum of quantized weights ($|\bar{\boldsymbol{W}}|$) and a constant ratio $\delta$.

$$\widetilde{\boldsymbol{W}} = N(0, \delta \cdot |\bar{\boldsymbol{W}}|_{max}) \tag{6}$$

## 3.3 TRAIN A WEIGHT QUANTIZED NEURAL NETWORK

---

**Algorithm 1** Training a $L$-layers quantization network:

---

**Input:** A mini-batch of inputs and targets($\boldsymbol{I}, \boldsymbol{Y}$), the pretrained full precision weights $\boldsymbol{W}$, $v$ distinguish quantized levels, learning rate $\eta$.
**Initialize:** quantizer $\Theta$ by Lloyd (1982), projection thresh $T$
**for** $i = 1$ **to** $L$ **do**
    quantized weight $\bar{\boldsymbol{W}}_i$ is calculated by $\Theta_i$
    noised weight $\hat{\boldsymbol{W}}_i = \bar{\boldsymbol{W}}_i + \widetilde{\boldsymbol{W}}_i$
**end for**
compute model output: $\hat{\boldsymbol{Y}} = forward(\boldsymbol{I}, \hat{\boldsymbol{W}})$
compute loss $\mathcal{L}$
**for** $i = L$ **to** $1$ **do**
    compute weight gradient $\dfrac{\partial \mathcal{L}}{\partial \boldsymbol{W}_i} = \dfrac{\partial \mathcal{L}}{\partial \hat{\boldsymbol{W}}_i}$
    update the full presicion weight $\boldsymbol{W}_i$ according to $\dfrac{\partial \mathcal{L}}{\partial \boldsymbol{W}_i}$ and $\eta$
    **if** $|\dfrac{\| \bar{\boldsymbol{W}}_i \odot \boldsymbol{W}_i \|_1}{\| \bar{\boldsymbol{W}}_i \odot \bar{\boldsymbol{W}}_i \|_1} - 1| > T$ **then**
        re-initialize $\Theta$ by Lloyd (1982)
    **end if**
**end for**

---

The training algorithm with quantization and noise awareness is shown in Algorithm 1. $\| \boldsymbol{X} \|_p = (\sum_i \sum_j | x_{ij} |^p)^{\frac{1}{p}}$ is its $p-$norm, $\boldsymbol{X} \odot \boldsymbol{Y}$ denotes the element-wise multiplication. There is one quantizer for each layer in the neural network. As some previous work do (Courbariaux et al., 2015; Rastegari et al., 2016), the quantized weights are used in the forward and backward propagations, but the update is applied on the full-precision weights. The Straight-Through-Estimator (STE) method (Bengio et al., 2013) are used during backward propagations. In order to reduce the frequency of optimizing the quantizer, we used the projection of the quantized weight vector on the full-precision weight vector to determine whether the quantizer need to be updated after each iteration. When the

projection exceeds a certain range, the quantizer will be updated. It is found that, if we start from a pre-trained neural network, the distribution of weights won't have too much change. This means the projection won't exceed the proper range during the whole training phase and the quantizer will be optimized only once at the beginning.

# 4  EXPERIMENTS

## 4.1  IMAGE CLASSIFICATION

In this section, we validate our quantization method on the CIFAR-10 (Krizhevsky et al., 2009) and the ImageNet-ILSVRC2012 (Russakovsky et al., 2015) dataset.

### 4.1.1  EVALUATION ON CIFAR-10

**ResNet-20:** We first evaluated the quantization method with ResNet-20 on the CIFAR-10 dataset, which consist of 50k training images and 10k test images in 10 classes. The data augmentation method is same as the previous work (He et al., 2016). For fine-tuning, we set the initial learning rate to 0.1, and scaled by 0.1, 0.1, 0.5 at epoch 80, 120, 160. The projection thresh $T$ was set to 1.5. We compared the performance of quantized model with several previous works, TTQ (Zhu et al., 2016), He (He & Fan, 2019), Lq-net (Zhang et al., 2018) and Li (Li et al., 2020). As shown in Table 1, our ternary model achieves 91.11% accuracy which is 0.42% lower than the full-precision model. Our 4-level model achieves 91.40% accuracy which is only 0.08% lower than the full-precision model.

Table 1: Classification Accuracy (%) of ResNet-20 architecture with 3-level and 4-level weights on CIFAR-10. "FP" denotes "Full Precision", "Quan" denotes "Quantization" and "Gap" means the accuracy of FP model minus the accuracy of Quan model.

| Method | Levels | Accuracy(%) | | |
|---|---|---|---|---|
| | | FP | Quan | Gap |
| TTQ (Zhu et al., 2016)[*] | 3 | 91.77 | 91.13 | 0.64 |
| He & Fan (2019) | 3 | 91.7 | 90.39 | 1.31 |
| **Ours** | 3 | 91.48 | 91.06 | 0.42 |
| **Ours w/ 1% noise** | 3 | 91.48 | 91.08±0.19 | 0.40±0.19 |
| Lq-net (Zhang et al., 2018)[*] | 4 | 91.6 | 90.2 | 1.4 |
| Li et al. (2020)[+] | 4 | 91.6 | 91.0 | 0.6 |
| **Ours** | 4 | 91.48 | 91.40 | 0.08 |
| **Ours w/ 1% noise** | 4 | 91.48 | 91.45±0.14 | 0.04±0.14 |

[*]Weights in first and last layer keep full precision. [+]Weights in first and last layer use 8-bit precision.

**VGG-like Network:** We also quantized VGG-like network on CIFAR-10 with ternary weights for evaluation. The model architecture and hyper-parameters are same as Hou's works (Hou & Kwok, 2018). The model has 6 convolutional layers and 2 fully-connected layers. Batch size was set to 50 and the learning rate starts 0.002 and decayed by a factor of 0.5 after every 15 epochs. The adam optimizer was used and the maximum number of epoch was set to 200. The projection thresh $T$ was set to 0.1.

We compared the classification accuracy of our method and several related methods (Table 2). The average accuracy of five trials using our method is higher than others. For DNNs which don't have pre-trained weights, training and quantizing it with our proposed method from scratch can also obtain good results. However, the quantizer may be updated frequently since the distribution of the weights will change a lot at the early stage of training. The closer to the solution, the more stable the weights distribution, that is, the fewer update times in the method.

Table 2: Classification Accuracy (%) of VGG-like architecture with 3-level weights on CIFAR-10 (FP:89.62).

| Method | Accuracy (%) |
|---|---|
| TWN (Li et al., 2016) | 89.36 |
| LAT (Hou & Kwok, 2018) | 89.62 |
| TTQ (Zhu et al., 2016) | 89.41 |
| **Ours (from scratch)** | $89.67 \pm 0.14$ |
| **Ours (fine tuning)** | $89.62 \pm 0.15$ |

### 4.1.2 EVALUATION ON IMAGENET

The ImageNet dataset consists of $1.2M$ training and $50K$ validation images. We conducted the experiment with the ResNet-18b and ResNet-50b (He et al., 2016). The pre-trained models from the PyTorch model zoo were used[1] . The data preprocessing is same as the origin paper (He et al., 2016). A $224 \times 224$ crop was randomly sampled from an image or its horizontal flip. Stochastic gradient descent (SGD) with the momentum of 0.9 was used to optimize weight parameters. The mini-batch was set to 64 and weight decay to 0.0001. The network was trained up to 90 epochs. The learning rate started from 1e-4, change to 2e-5, 4e-6 at epoch 30, 40 correspondingly, and then scaled by 0.5 after every 5 epochs. The projection thresh $T$ was set to 0.1.

To make more generalization, we quantized the weights in all the layers of a DNN, including the first and the last layer in the experiments. For a fair comparison with some previous work, we also conducted the experiments that did not quantize the first and last layer.

When the precision is up to 8-level, both the top-1 and top-5 accuracy $(70.0\%/89.3\%)$ outperform the full precision model $(69.8\%/89.1\%)$. Similar results were obtained in ResNet-50b network. The top-1 accuracy of ResNet-50b model with 4-level precision is only $0.3\%$ lower than the full-precision model. We compared our methods with some previous studies, which were also based on the ResNet architecture. The experimental results are listed in Table 3. In most experiments, our results achieved the least classification accuracy gap.

Table 3: Comparison the validation Top-1 and Top-5 accuracies gap (%) between ResNet-18b and ResNet-50b with quantized weights or float weights using various quantization methods on ImageNet. The baselines of these two full precision models are 69.8/89.1 and 76.1/92.9.

| Method | weight levels of ResNet-18b | | | weight levels of ResNet-50b | | |
|---|---|---|---|---|---|---|
| | 3 | 4 | 8 | 3 | 4 | 8 |
| TTQ-B*(Zhu et al., 2016) | 3.0/2.0 | - | - | - | - | - |
| ABCnet*(Lin et al., 2017) | - | 5.6/4.0 | - | - | - | - |
| Lq-net*(Zhang et al., 2018) | - | 2.3/1.5 | 1.0/0.7 | - | 1.3/0.9 | - |
| Jung*(Jung et al., 2019) | - | - | 0.3/0.3 | - | - | - |
| He & Fan (2019)* | 1.7/1.2 | - | - | - | - | - |
| He & Fan (2019) | 3.9/2.4 | - | - | 2.1/1.2 | - | - |
| Yang et al. (2019)* | 1.2/0.6 | - | -0.1/-0.1 | 1.2/0.6 | - | 0.2/0 |
| **Ours*** | 1.2/0.7 | - | - | **0.8/0.4** | - | - |
| **Ours** | 2.5/1.4 | **1.3/0.6** | **-0.2/-0.2** | 1.3/0.7 | **0.3/0.2** | **-0.3/-0.3** |

*Weights in first and last layer keep full precision.

We searched the relative noise $\delta$ during training the ResNet-18b with ternary weights. No significant difference was involved when $\delta$ ranges from 1% to 4%, so we used 2% in all following experiments. The comparisons of the inference performance between our quantized models with weight noise and previous models without weight noise are shown in Figure 3.

---

[1]https://pytorch.org/docs/stable/_modules/torchvision/models/resnet.html

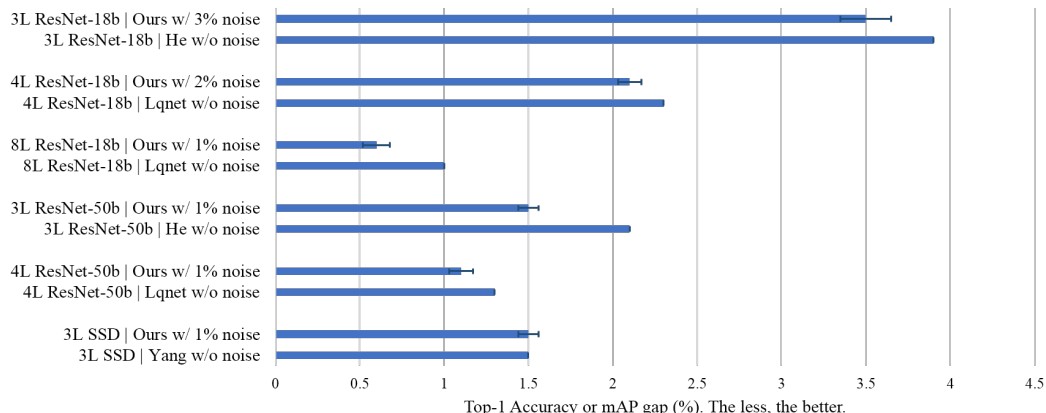

Figure 3: Quantization performance comparison between our quantized model with weight noise and others' model without weight noise.

## 4.2 Object Detection

In this section, we evaluated our proposed approach on a general object detection task. Our experiments were conducted on SSD (single shot multibox detection) architecture (Liu et al., 2016). All methods were fine-tuned on the same pre-trained VGG16 network. The models are trained on the PASCAL VOC 2007 and 2012 training datasets and tested on the PASCAL VOC 2007 test dataset. For a fair comparison, except for the final convolutional layers with $1 \times 1$ kernels and the first convolution layer, the parameters of all other layers in the backbone VGG16 are quantized.

The input images were resized to $300 \times 300$. SGD optimizer with the momentum set to 0.9 was used. We used the $1e^{-3}$ learning rate for 80k iterations, then continue training for 20k iterations with $1e^{-4}$ and $1e^{-5}$. The batch size was set to 16, and the weight decay was $5e^{-4}$. The results are shown in Table 4. When our ternary weights was injected with 1% noise, the mAP gap ($1.5\%\pm 0.06\%$) of our method was still not less than ADMM (Leng et al., 2018)'s and Yang et al. (2019)' without noise. Our 8-level model performed better than full-precision model.

Table 4: mAP(%) of SSD on PASCAL VOC object detection. The performance of the full-precision model is 77.8.

| Methods | Weight levels | | |
|---|---|---|---|
| | 3 | 4 | 8 |
| ADMM | 76.2 | - | 77.6 |
| Yang | 76.3 | - | 77.7 |
| **Ours** | 76.7 | 77.1 | 77.9 |
| **Ours w/ 1% noise** | 76.3±0.06 | 77.1±0.03 | 78.0±0.08 |

## 4.3 Semantic segmentation

We evaluated our method on the PASCAL VOC 2012 segmentation dataset (Everingham et al., 2010) and used the PSPNet (pyramid scene parsing network) architecture. Following the same settings in Zhao et al. (2017), we used augmented data with the annotation of Hariharan et al. (2011) resulting 10,582, 1,449 and 1,456 images for training, validation and testing. The backbone of our model was ResNet50. We evaluated the model with several-scale input and used the average results. The batch size was set to 8 constrained by limited memory. The learning rate was initialized by $1e^{-2}$ and decayed by power policy with 0.9. SGD optimizer with the momentum set to 0.9 was used. Our 8-level model can achieve the same performance with full-precision model 5.

Table 5: Average IoU (mIoU%) and pixel Accuracy (pAcc%) of PSPNet on PASCAL VOC 2012 valid set with our proposed method.

| Weight levels | mIou | pAcc |
|---|---|---|
| 3 | 75.5 | 93.6 |
| 4 | 76.3 | 93.9 |
| 8 | 77.3 | 93.9 |
| float | 77.3 | 94.2 |

### 4.4 PERFORMANCE ON REAL DEVICES

To further demonstrate the reliability of the results, we map the weights to the real memristors and use the measurement conductance to verify our method. The conductance ($\mu S$) range of our memristor device is [2.1, 17.85]. To represent the negative weights, the differential conductance ($G$) of two devices is used. $G$ is given by $G = G^+ - G^-$, where $G^+$ ($G^-$) is the conductance of a positive (negative) device. To reduce the mapping overhead, we used a unified reference conductance ($G_{ref}$) in all differential fairs. To represent a positive (negative) weight, the negative (positive) device is mapped with $G_{ref}$. For one layer's quantized weights in DNN model, we firstly normalized them by dividing with the maximum value of absolute weights. Then we mapped the normalized weights to $G$ by multiplying with $15.75\mu S$ and used $2.1\mu S$ as $G_{ref}$.

Taking an example of a well-trained ResNet-18b model with 4-level weight, the set of quanta in the first convolutional layer is [-0.25, -0.01, 0.11, 0.38]. After scaling the weight set, we get the $G$ ($\mu S$) set which is [-10.36, -0.57, 4.52, 15.75]. The $G^+$ ($\mu S$) set is [2.1, 2.1, 6.62, 17.85], the $G^-$ ($\mu S$) set is [12.46, 2.67, 2.1, 2.1]. The standard deviation of the measured $G$ is $0.32\mu S$, which is approximately equal to $0.02 * |G|_{max}$. Figure 4(a) shows the simulated and measured distribution of the differential conductance. The measured and simulated differential conductance have the same statistical property. As shown in Figure 4(b), we compared the inference performance of three types of weights: A) simulated weights training with NATS, B) measured weights training with NATS, C) measured weights training without NATS, using 4L, 8L ResNet-18b and SSD model. The results of type A and type B are very close. The results of type B are better than those of type C indicating that NATS improved the robustness of the DNN model.

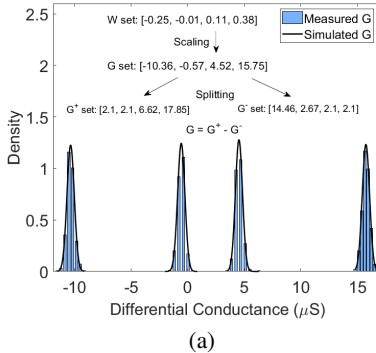

(a)

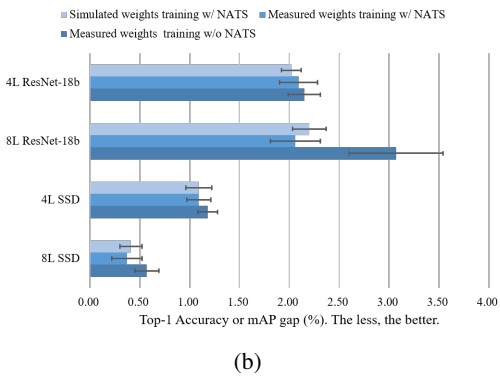

(b)

Figure 4: (a) shows the simulated and measured distribution of differential conductance ($\mu S$) which mapped from the 4L weight of the first convolutional layer in ResNet-18b. (b) shows the top-1 accuracy gap (%) of ResNet-18b and mAP gap (%) of SSD model using 4L, 8L weights with simulated or measured noise.

## 5 Discussion

### 5.1 Generalized quantization method in digital systems

A generalized quantizer can obtain less quantization error than a uniform one in theory when the data distribution is non-uniform. However, in digital computers, it needs more memory or additional operations to process a set of non-uniform data. As shown in Figure 5, a series of data are quantized to $\{1, 2, 4, 6\}$. In a digital system, it will use 3 bits to store each number with binary code. Although we can store these numbers also with 2 bits per number, a mapping function, that is '00' = 1, '01' = 2, '10'= 4, '11' = 6 must be stored and called whenever using these numbers. This additional cost limits the application of the generalized method in digital systems to a certain extent. However, in the CACIM systems, the data is stored in analog quantities, and the operation is based on the analog computing scheme. No matter what the exact value is, there is no significant difference in the storing or computing operation. This is why the generalized quantization method is more suitable for the CACIM system.

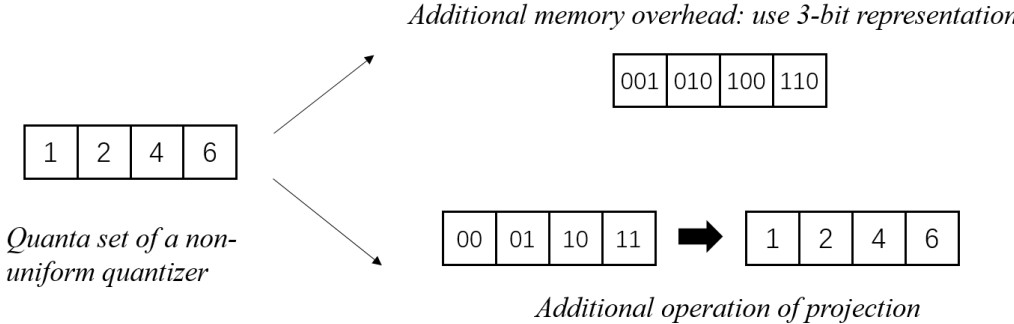

Figure 5: Processing of a series of non-uniform data in digital computers. Additional memory overhead or additional operation of projection is needed.

### 5.2 Principles of CACIM-oriented algorithm design

Different from the digital system, the CACIM system is mainly based on analog computing, which may introduce a great difference when designing and using neural network algorithms. For the quantization methods, the analog computing scheme means that the weights are represented and calculated on real quantities, not binary numbers implemented by high and low levels in the digital system. No matter what the exact weight is, the read or the computing is in the same process, that is, apply a voltage and collect the current. Therefore, we can utilize this characteristic to select a more powerful quantization algorithm, such as the GDQ, which may obtain non-uniform results. Besides the quantization, the characteristics of analog computing may also play important roles in other scenarios. Since the current is directly accumulated and collected along the column of the crossbar, the behavior of each device at the column may influence the results. If we ignore these behaviors when designing the algorithm, they will degrade the performance of the algorithms. That is why we usually call them non-ideal characteristics. However, if we have a clear understanding of these characteristics, we can overcome them. Such as we used the noise-aware training scheme to improve the accuracies in this work. Furthermore, we can even utilize them to achieve better results, such as using the variation of the device to provide the stochasticity needed in the algorithm (Lin et al., 2018; 2019), using the I-V nonlinearity of the device to introduce the nonlinearity and displace the activation functions, or using the relaxation behavior of the device to efficiently implement the weight decay operation in the training process. As demonstrated in this work, the understanding of the hardware system is helpful for us to design better algorithms.

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
