# OpenReview forum: "Improving the accuracy of neural networks in analog computing-in-memory systems by a generalized quantization method"
_ICLR.cc/2021/Conference — Reject_

### Official Review · AnonReviewer4 · 2020-10-24
**Less Innovative Quantization Method on a new Hardware System**

**Rating:** 5
**Confidence:** 4

**Review:**

Starting from the property of CACIM, this paper found that weight quantization fits CACIM's analog calculation process and proposed a general quantization method. Specially, it use Lloyd;s quantizer to quantize weights and followed the commonly used quantization training method (STE) to update the full-precision weights.

Pros:
1. Find a good usage of model quantization in a new hardware system.

Cons:
1. The biggest problem from my perspective is that: the proposed quantization method is disentangled with CACIM: It only considers that quantized weight has its unique advantages in analog computation. But CACIM is not involoved in how to acchieve the quantized weight. Other quantization method can also be applied to CACIM. Although experiments show outperformance over certain baselines, it only shows that the proposed method can achieve satisfying quantization. Overall, property of CACIM does not contribute to the design of quantization method.
2. Besides, the proposed quantization method is not innovative. Lloyd's formula has been used in quantization before [1][2]. Not to mention the noise-aware training and STE.
3. This paper's focus is on CACIM, but experiments are not conducted on this hardware. It will be much convincing if the proposed method is applied on CACIM.
4. Is $|\frac{\bar{W}_i W_i}{\bar{W}_i \bar{W}_i} - 1| > T$ in Algorithm 1 typo?

[1] https://github.com/spencerkent/generalized-lloyd-quantization
[2] https://openreview.net/forum?id=rJ8uNptgl

---

> ### Author Response · Authors · 2020-11-23
> **To Reviewer #4**
>
> Thank you for the positive review and for taking the time to thoroughly read and comment on our paper.
> 1.	We don’t agree that CACIM is not involved in how to achieve the quantized weight. Analog weights and no weights data transfer are characteristics of CACIM. There are no weights encoding and decoding process in CACIM. That is the key characteristics we utilized. When the number of distinct levels of memristor devices is limited, our proposed method results in better performance than other quantization methods.
> 2.	The Lloyd’s formula is not what we proposed and is what we adopted. In reference [2], the authors proposed hessian-weighted k-means clustering and entropy-constrained scalar quantization.
> Although plenty of quantization methods had been proposed. The results in this work show that utilizing the analog characteristic of CACIM systems can obtain better quantization. That is the main contribution of this work. Here we added same explanation in the Part 5.2 in the new submission. We simulated the DNN inference on CACIM by adding noise to quantized weights. This is the same way of DNN inference on CACIM.  More experimental results are added in the new submission.
> 3.	The experiments conducted on the CACIM system are important. We conducted the experiments in the last weeks and showed the results in Part 4.4 in the new submission.
> 4.	Sorry, it’s our mistake. We firstly define two notations. $\parallel {{X}} \parallel_{p} = (\sum_{i}\sum_{j}\mid x_{ij}\mid^p)^{\frac{1}{p}}$ is its $p-$norm. ${X}\bigodot {Y}$ denotes the element-wise multiplication.
> Then the condition is $|\dfrac{ \parallel {\bar{W_{i}}}\bigodot {W_{i}} \parallel_{1} } {\parallel  {\bar{W_{i}}}\bigodot {\bar{W_{i}}} \parallel_{1}} - 1|> {T}$.
> We revised the mistake in the new submission.

---

### Official Review · AnonReviewer3 · 2020-10-27
**Quantization to enable DNN deployment onto analog crossbars**

**Rating:** 3
**Confidence:** 5

**Review:**

In order to improve robustness of analog written weights to circuit variation, quantization is used. My first question is why is that the case? The claim that quantization reduces analog noise does not seem to be correct. As far as I can tell, this setup leads to quantization noise on top of analog noise. This is a very serious point since the authors just assume this claim to be true (first para of page 3) and that is the whole motivation. The claim should be justified, or at least a reference should be provided. P.S., I do not think the claim is correct; two noise sources (quantization + analog) is worse than one noise source (analog only).

Second issue: what is the contribution of this paper? As far as I can tell, two old methods (LLoyd max and dithering) are used and that's it. Sure, the experimental section is massive but this paper does not claim to be a pure empirical study. In fact, the claim is that for the first time, a general quantizer is used. Those methods have been around for decades!

The experiments only use quantization. One would expect that CACIM would be emulated somehow and we would see that indeed the proposed method improves CACIM's accuracy (which I don't think it would as discussed above). Even the conclusion talks about digital systems. So CACIM was (erroneously) used to motivate the work but then completely forgotten.

Post Rebuttal Comments:

I thank the authors for their feedback. I have no modification to make to my original review.

---

> ### Author Response · Authors · 2020-11-23
> **To Reviewer #3**
>
> We want to thank the reviewer for their careful reading and providing a lot of critical comments! Below we address the concerns mentioned in the review:
> 1.	We agree that two noise sources are involved. In order to improve the robustness of analog written weights to circuit variation, a noise-aware training scheme is used. To apply a DNN model in CACIM systems, we usually train a model on GPU or CPU and then mapping the weights on crossbar arrays in CACIM systems. Two main issues are involved. Firstly, the mapping overhead is directly related to the precision of weights. We used quantized weights to decrease mapping overhead. Secondly, the mapping noise is non-negligible and degraded the performance of networks severely. We used a noise-aware training scheme to improve the robustness to noise. Both the quantization method and the noise-aware training need to tune the weights. Therefore, in this paper, we combine the quantization method and the noise-aware training to obtain the low-precision and high-robustness weights.
> 2.	To quantize DNN models in CACIM systems, the previous works even the newest ones [1],[2] used uniform or log-wise quantization algorithm. Their performance is not as good as ours. It is all because we found that there are no weights encoding and decoding processes in CACIM systems and utilized this characteristic to select a more powerful quantization algorithm. This characteristic is so important that not only allows us to use more powerful quantization algorithms but may also design some other specific algorithm for CACIM systems to improve the performance. We discussed the principles of CACIM-oriented algorithm design and explained why we used the generalized quantization method on the last page of the new submission.
> 3.	By adding random noise to each weight in the inference phase, we tried to simulate the performance of a DNN in the CACIM system. Table 1-4 in the paper included the experiment results which are obtained in this simulation way. And to further demonstrate the reliability of the results, we conducted more experiments that mapping the weights to the real crossbar arrays and using the measurement values to verify our method. Related experiment results are shown in Part 4.4 in the new submission. The noise-aware training scheme indeed improved the robustness of the DNN model.
>
> [1] Qu, Songyun, et al. "RaQu: An automatic high-utilization CNN quantization and mapping framework for general-purpose RRAM Accelerator." 2020 57th ACM/IEEE Design Automation Conference (DAC). IEEE, 2020.
>
> [2] Cai, Yi, et al. "Low bit-width convolutional neural network on rram." IEEE Transactions on Computer-Aided Design of Integrated Circuits and Systems (2019).

---

### Official Review · AnonReviewer1 · 2020-10-28
**A new non-uniform weight quantization method. Conenction to Compute-in-Memory system not strong. Comparison with one similar method is missing.**

**Rating:** 5
**Confidence:** 4

**Review:**

Summary:
This paper proposes a method to train weight-quantized neural networks.  The authors propose to directly calculate the endpoints that minimize the quantization error according to the weight distribution of each layer. Empirical results on image classification tasks and object detection tasks show that the proposed method outperforms other compared weight quantization methods under the same number of bits.

Strengths:
- The paper is clearly written and the proposed method is simple.
- Experiments are performed on image classification tasks CIFAR-10 and ImageNet, with extensive comparisons with other methods.

Weaknesses:
- Since the Floyd algorithm requires alternating minimization between q and e, one concern is that this may be costly. How often is the Floyd algorithm called in Algorithm 1? What is the training time compared with other methods?

- Comparison with one important reference [1] is missing. [1] also learns the quantized values, but (i) through the step size instead of the endpoints,  and (ii) uses uniform distributed quantized values instead of non-uniformly distributed ones. However, the reported results in [1] show that their 3-bit, 4-bit and 8-bit quantized models on ImageNet have  smaller accuracy gaps with the full-precision baseline compared to the proposed method.

- One other concern is that while the authors claim that this method is proposed for compute-in-memory (CIM) system, and discussed in section 5 that the property of the CIM does not restrict the quantized values to be uniformly strictly, compared to digital systems, there is no empirical comparison of the efficiency (e.g. storage, latency) of the proposed method and other quantization methods in CIM. Do they just perform similarly in terms of computation and memory efficiency? If so, when the uniform quantization in [1] already has good accuracy, using generalized quantization as in the proposed method does not seem quite well-motivated?

[1] Esser, Steven K., et al. "LEARNED STEP SIZE QUANTIZATION." International Conference on Learning Representations. 2019.

---

> ### Author Response · Authors · 2020-11-23
> **To Reviewer #1**
>
> Thanks for your time in reviewing our work and pointing out our potential issues.
> 1.	The complexity of Lloyd’s algorithm is O(N⋅e^(-t) ), where N is the number of distinct quantization levels and t is the termination threshold. The algorithm will only be used when the weight projection falls outside a given range. In a typical fine-tuning process, the Lloyds’ algorithm will only be triggered a few times. For example, when fine-tuning a Resnet-18b up to 45x10^4 iterations on ImageNet task from a pre-trained model, the Lloyds’ algorithm is only used less than 10 times, which is negligible. We compare the training time of ternary weight ResNet-18 model with that of full precision one, the former is empirically only 1.064 times than the latter. The additional training time cost introduced by quantization is acceptable.
> 2.	There are two main differences between the reference [1] and our work. Firstly, the quantization precision is different. Reference [1] used 3-bit, 4-bit, and 8-bit quantized models. While we used 3-level, 4-level, and 8-level quantized weights. 3-bit is 8-level. Secondly, both the weights and activations are quantized in reference [1], while only the weights are quantized in our work. In CACIM systems, the weights are represented by the conductance of analog devices. The multiply-accumulate operation directly used the analog conductance without converting to digital signals. While the activations are usually implemented by digit circuits. The difference between the underlying representation mechanisms let us decide not to quantize the weights and activations together.
> The experimental results show that our 4-level and 8-level quantized models on ImageNet have smaller accuracy gaps with the full-precision baseline compared to the 2-bit and 3-bit quantized model in reference [1]. Note that the reference used 8-bit weights and activations in the first and last layers, which can provide a large benefit to performance. In our work, we quantized all the layers with the same precision.
> 3.	In the inference phase of DNN on the CACIM system, our proposed method and other quantization methods in CACIM just perform similarly in terms of computation and memory efficiency. After different quantized weights obtained by different quantization algorithm mapping to memristor devices in the CACIM system, the hardware computing process is independent of the conductances of memristor devices. We explained why we used the generalized quantization method in the second subsection of Part 5 in the new submission.

---

### Official Review · AnonReviewer2 · 2020-10-30
**Interesting paper but Limited Novelty**

**Rating:** 4
**Confidence:** 4

**Review:**

Contribution:
1. Using quantization to make the deployment of CACIM feasible is straightforward and interesting.

2. The proposed method was validated on CIFAR-10 and ImageNet with different models.


Cons:
1. Combining quantization and CACIM is interesting. However, I am concerning the novelty of methods used in this paper. The 'general quantization' used in this paper is quite similar to Product quantization. [Stock, Pierre, et al. "And the bit goes down: Revisiting the quantization of neural networks." arXiv preprint arXiv:1907.05686 (2019).] Quantization by grouping and k-means is well-known in this community.

2. In the abstract, the authors claim that 'The analog weight has its unique advantages when doing quantization. Because there is no encoding and decoding process, the range of quantization function will not affect the computing process.' When we do quantization for the digital device, we quantize (almost) continual (high precision) numbers into discrete numbers. The range of quantizer is telling us which range we want to pay more attention to during quantization. Although I agree with the claim that 'there is no encoding and decoding.', I didn't follow the sentence 'the range of quantization function will not affect the computing process'. More explain should be provided about 'what is the definition of the range of quantizer and why it will impact quantization in the digital system and it will not impact quantization in the analogy system.'

---

> ### Author Response · Authors · 2020-11-23
> **To Reviewer #2**
>
> We appreciate the reviewer for your kind review and suggestions.
> 1. About the concerns about the novelty of our methods, we found that there are no weights encoding and decoding process in the CACIM system, which may be very suitable for the generalized quantization method. According to the analysis and simulation results in the paper, we proved that the generalized quantization method can improve the accuracy with almost the same inference cost (same quantization levels). As far as we know, there is no previous study that has recognized this characteristic of the CACIM system and takes advantage of it to improve the performance. We discussed the principles of CACIM-oriented algorithm design on the last page of the new submission.
>
> 2. “The range of quantizer” in the paper means the set of quanta. As discussed in Section 5 of the paper, if we consider two sets of quanta: A-{1,2,3,4}, B-{1,2,4,6}. In digital computers, set B needs additional memory overhead of additional operation projection during the computing process. Which in CACIM, the quanta are storage by analog devices. Set A and set B have the same quantization levels so that the mapping overhead and inference time cost are same. The statement of the range of quantizer may be misleading. We updated the statement in the modified version.

---

### Decision · Program_Chairs · 2021-01-07
**Final Decision**

**Decision:**

Reject

**Comment:**

This work develops a weight-quantization method for deep neural networks that is suitable for a type of analog hardware system known as crossbar-enabled analog computing-in-memory (CACIM).  The goal of this work is to train models on GPUs in such a way that they retain their predictive accuracy during inference when deployed on the analogue hardware system.

Pros:
* Good adaptation of quantization methods to the CACIM system
* Simple method
* Validation of the proposed method on multiple datasets and models

Cons:
* Lack of novelty: the proposed method is a simple combination of two popular methods, LLoyd's quantization and noise-aware training

All reviewers appreciate the simplicity of the method and the good fit to the hardware.  The authors responded to all reviews and two reviewers acknowledged the authors' response.  The authors acknowledge some reviewer observations (motivation of quantization as reducing analogue noise, lack of experiments on the actual CACIM system), and the authors added an experimental evaluation on the actual physical CACIM system showing that their method performs well.

Overall the work is well-executed and the proposed method is a good fit to the CACIM system.  However, the proposed quantization method is a straightforward adaptation of popular quantization methods.